

# RUMD: A general purpose molecular dynamics package optimized to utilize GPU hardware down to a few thousand particles

Nicholas P. Bailey[1⋆], Trond S. Ingebrigtsen[1], Jesper Schmidt Hansen[1], Arno A. Veldhorst[1], Lasse Bøhling[1], Claire A. Lemarchand[1], Andreas E. Olsen[1], Andreas K. Bacher[1], Lorenzo Costigliola[1], Ulf R. Pedersen[1], Heine Larsen[1], Jeppe C. Dyre[1] and Thomas B. Schrøder[1♯]

**1** "Glass and Time", IMFUFA, Dept. of Science and Environment, Roskilde University, Roskilde, Denmark

⋆ nbailey@ruc.dk, ♯ tbs@ruc.dk

## Abstract

RUMD is a general purpose, high-performance molecular dynamics (MD) simulation package running on graphical processing units (GPU's). RUMD addresses the challenge of utilizing the many-core nature of modern GPU hardware when simulating small to medium system sizes (roughly from a few thousand up to hundred thousand particles). It has a performance that is comparable to other GPU-MD codes at large system sizes and substantially better at smaller sizes. RUMD is open-source and consists of a library written in C++ and the CUDA extension to C, an easy-to-use Python interface, and a set of tools for set-up and post-simulation data analysis. The paper describes RUMD's main features, optimizations and performance benchmarks.

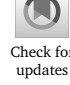
## Contents



## 1   Introduction

This paper describes the Roskilde University Molecular Dynamics (RUMD) package. RUMD is a molecular dynamics [1, 2] (MD) code running on Graphical Processing Units (GPU's) from NVIDIA. RUMD was developed to achieve good performance at small and medium system sizes, while remaining competitive with other GPU-MD codes at large sizes. The attention paid to small sizes distinguishes RUMD from many other GPU-MD codes. It has been in development since 2008, and available as open-source software,[1] since 2011. The newest version 3.3, was released in June 2017.

The rise of GPU-based computation has been discussed by various authors [3–9]. MD is a good candidate for GPU-acceleration, as discussed by Stone *et al.* [5], since it involves a reasonably high arithmetic intensity, that is number of floating point operations per memory access. Several groups have developed MD codes based on GPUs from scratch or incorporated GPU-acceleration into existing projects. Examples of the former include HOOMD-Blue [10–13], ACEMD [14], OpenMM [15, 16] and HAL's MD [17] while the latter include NAMD [18], LAMMPS [19], AMBER [20, 21] and Gromacs [22] and GENESIS [23]. Other works involving GPU-based MD codes, going back to 2007, can be found in Refs. [24–35]. We omit a detailed exposition of GPU programming basics here; for a good overview of massive multi-threading using CUDA see the relevant section in the article by Anderson *et al.* [10]. For further information the reader can consult the book by Kirk and Hwu [36] as well as the CUDA programming guide [37].

The large computational power of modern GPUs comes primarily from the large number of hardware cores, each executing a number of software threads. Here we focus on two of the most recent architectures, Kepler (2012) [38] and Pascal (2016) [39]. As an example, the GeForce Gtx780Ti card (Kepler architecture) has 2880 cores and a theoretical single-precision peak-performance of 5.0 TFlops ($5 \times 10^{12}$ floating point operations per second). A key element to achieve good performance from a GPU is that the number of active software threads should be much larger than the number of hardware cores in order to hide latency of memory access. This makes it a challenge to utilize the GPU hardware when the number of particles $N$ is relatively small ($N \sim 10^3 - 10^4$). The obvious choice for parallelization, namely having one thread compute the forces for one particle, is clearly not efficient when the optimal number of threads exceeds the number of particles. There are three reasons to focus on utilizing the GPU hardware even at small system sizes; i) Simulating long time scales rather than large systems. This is of interest, for example, in the field of glass-forming liquids. Here a system size of $10^4$ particles is considered large, but the interest is in studying as long time scales as possible. Note that finite size effects are relatively limited in these systems; for example Karmakar, Dasgupta, and Sastry [40] found convergence of diffusivity and relaxation time for a standard model glass-former already at N=1000. ii) As a building block for multi-GPU simulations [41] (RUMD currently uses one GPU per simulation). If one wants to simulate,

---

[1]RUMD software is freely available at http://rumd.org.

```
# Import RUMD
import rumd
from rumd.Simulation import Simulation

# Create a simulation object, and import an initial configuration.
sim = Simulation("start.xyz.gz")

# Create a pair potential and associate it with the simulation object
pot = rumd.Pot_LJ_12_6(cutoff_method=rumd.ShiftedForce)
pot.SetParams(i=0, j=0, Sigma=1.0, Epsilon=1.0, Rcut=2.5)
sim.SetPotential(pot)

# Create an integrator and associate it with the simulation object
itg = rumd.IntegratorNVT(timeStep=0.004, targetTemperature=1.0)
sim.SetIntegrator(itg)

# Run a simulation. Data are saved on disk, can be analyzed by various tools
sim.Run(1000000)
```

Figure 1: Script showing the Python code needed to run a simple simulation, in this case a single-component Lennard-Jones fluid simulated at constant temperature 1.0 for one million time steps of size 0.004 (Lennard-Jones units). The number of particles and the density is determined by the initial configuration contained in the file start.xyz.gz

say, $10^5$ particles using 10 GPU's, the single-GPU performance obviously needs to be good for $10^4$ particles. iii) Much of the future development in GPU and other many-core hardware will probably be in increasing the number of physical cores. Thus, what might today be considered a large system, might in the future be considered a small/medium sized system where special care needs to be taken to utilize the GPU hardware. To optimize the use of the hardware RUMD allows multiple threads per particle; this approach has also been considered in two recent publications [42, 43]. Finally, we note a very recent paper describing the use of large ensembles of MD simulations of small systems [44], an approach which would also be very much suited to running on GPUs.

The paper is organized as follows. Section 2 contains a brief overview of RUMD's features. The main part of the paper focuses on the methods used for calculating the non-bonding pair interactions and the generation of the neighbor-list. These are the most computationally demanding parts of an MD simulation and where RUMD distinguishes itself from most other GPU-MD codes. Section 3 discusses the challenges of utilizing the GPU hardware at small system sizes, and section 4 gives an overview of the optimization strategies employed in RUMD. Section 5 describes the calculation of non-bonding pair-forces, while sections 6 and 7 describe two different methods for generating the neighbor-list. Section 8 provides benchmarks of RUMD in comparison to three different GPU extensions of LAMMPS [19], as well as an analysis of the effect of the different optimizations employed in RUMD. Section 9 describes RUMD's performance for electrostatic (Coulomb) interactions. Section 10 provides a short summary.

## 2 RUMD: Features

Below we list the main features of RUMD; for more information please see the tutorial and user manual included with the software and available from the project's website `rumd.org`.

**Python interface:** The user controls the software via a Python interface which allows simulations of considerable complexity to be implemented straightforwardly. An example of a simple user Python-script is given in Fig. 1.

**Pair potentials:** 12-6 Lennard-Jones, generalized Lennard-Jones, inverse power law, Gaussian core, Buckingham, Dzugotov, Girifalco, Yukawa, and more. New pair potentials are easily added, as described in the tutorial. Three different "cutoff methods" for truncating the pair potential are provided: simple truncation with no shift, truncation plus shift of the potential energy to ensure continuity, and truncation plus shift of the pair force [45] to ensure its continuity (this corresponds to adding a linear term in the potential).

**Other interactions:** Intramolecular interactions, including both rigid-body constraints [46, 47] and flexible terms—bond-stretching forces, angular forces and dihedral forces. Three-body angle-dependent non-bonding interactions [48]. Many-body interactions for metals based on effective medium theory [49].

**Integrators:** NVE (Verlet/Leap-frog), NVT (Nosé-Hoover), NPT [50], NVU (geodesics on the constant potential energy surface) [51, 52]. Couette shear flow using the SLLOD equations of motion and Lees-Edwards boundary conditions.

**File formats:** Configurations are stored in the xyz format with extensions, compressed using gzip; data can be saved block-wise logarithmically in time for efficient use of disk space while allowing the study of a large range of time scales in a single simulation; molecular structure (bonds, angles and dihedrals) is specified in separate topology files. Tools for creating initial configurations and topology files are provided.

**Analysis tools:** Basic statistics of energy, pressure, etc. for thermodynamics. Measures of structure: radial distribution function, static structure factor, radius of gyration, mean-square end-to-end distance. Measures of dynamics: mean-square displacement, incoherent intermediate scattering function, non-Gaussian parameter, end-to-end vector autocorrelation function, Rouse-mode autocorrelation function. New analysis tools are added regularly. Analysis tools work on data stored during simulations and can be applied at the end of or during a simulation. The user may define customized on-the-fly analysis tools written in Python.

**Autotuner:** A script for optimizing internal parameters—specifically, the choice of algorithm for generating the neighbor list, the neighbor-list skin size, and the way the generation of the neighbor list and the calculation of non-bonding forces are distributed among the GPU threads. The autotuner is described in the appendix.

RUMD is mostly implemented in single precision. This leads to a drift in the total energy when running long constant-energy (NVE) simulations, but is not an issue for NVT and NPT simulations where a thermostat is applied. RUMD can be made fully double precision by a search and replace in the source code - we are planning to implement a more elegant way for the user to choose between single and double precision. RUMD uses a single GPU per simulation; support for multiple GPU simulations is planned for future development.

## 3 The challenge of utilizing the GPU at small system sizes

Consider NVIDIA's Kepler GK110 architecture that appeared in 2013. One of the Kepler design goals was power efficiency, which was partly achieved by increasing the number of cores while decreasing the clock speed compared to the previous Fermi architecture. Thus each streaming multiprocessor (of type SMX) has 192 cores, and the GPU has up to 15 SMX units. The GTX 780Ti card contains the maximum 15 SMX units, giving 2880 cores. Furthermore, the number of software threads needs to be much larger than the number of physical cores, in order to hide

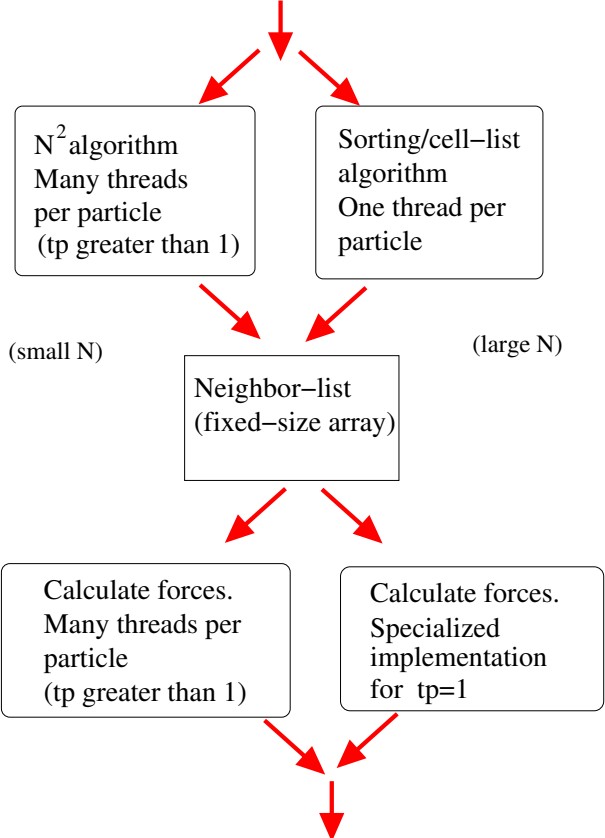

Figure 2: Schematic diagram representation of the two algorithms for neighbor-list generation, and the force calculation algorithm. The latter uses multiple threads per particle (tp), but an implementation also exists for the special case tp = 1.

memory access latency. This poses a challenge when small systems of the order of thousands of particles are concerned. In order to use as many threads as possible, one must therefore have multiple threads computing the force on one particle.

Having multiple threads per particles entails some overhead, in particular the summing of the force contributions over the threads allocated to a given particle. This means that as the system size increases, it becomes less useful to have more than one thread per particle. We control this by the parameter $t_p$ (threads per particle, denoted `TPerPart` in the code), and let the autotuner pick the optimal value for a given simulation. The optimal value of $t_p$ depends primarily on the number of particles, but also on density and the range of the potential. We use a separate kernel involving a single thread per particle for larger sizes (see Fig. 2); this is faster than setting $t_p = 1$ in the general kernel.

Rovigatti *et al.* [53] have recently discussed the possible advantages of "vertex-based" (atom-decomposition [54], one thread per particle) versus "edge-based" (force-decomposition [54], one thread per interaction) parallelism. Our approach includes the former and a range of intermediate cases, while not taking it to the extreme of one thread per interaction.

## 4 Optimization strategies used in RUMD

As in any general purpose MD software a data structure to keep track of neighbors for the non-bonding pair interactions is necessary to reduce the complexity of the force calculation

from $O(N^2)$ to $O(N)$ [1,2]. We use a classical Verlet-type neighbor list, stored as 2-dimensional fixed-size array of size $Nn_{\max}$ where $n_{\max}$ is the assumed maximum number of neighbors per particle. If this happens to be exceeded the neighbor-list is automatically re-allocated with doubled capacity. For smaller systems we set $n_{\max} = N$ from the start to avoid the overhead of checking for overflow. Neighbors within $r_c + s$ are listed, where $r_c$ is the maximum cut-off associated with the potential, and $s$ is the extra skin included so that the neighbor-list does not need to be rebuilt every step. The optimal value of the skin is determined by the autotuner.

We now describe the methods employed in the calculation of short-range non-bonding forces and the generation of the neighbor-list. The main four optimizations are as follows:

1. Multiple threads per particle ($t_p \geq 1$) in force calculation and neighbor-list generation. The autotuner chooses the best value for $t_p$.

2. Two methods for rebuilding the neighbor-list: $O(N^2)$ method ($t_p \geq 1$) for small system sizes, and an $O(N)$ method ($t_p = 1$) for larger sizes. The autotuner picks the best method.

3. Use of the so-called "read only data-cache" for reading positions (for NVIDIA devices of so-called "compute capability" [37] at least 3.5 this can be done straightforwardly via the function `__ldg()`).

4. Use of pre-fetching when reading from the neighbor-list to compensate for memory access latency.

# 5   Force calculation

The force calculation kernel (routine executed on the GPU) is shown in Fig. 3. Short-hand notation for common quantities used in this and the following CUDA-kernels are given in Table 1. The force kernel uses in general $t_p \geq 1$, although a separate implementation for $t_p = 1$ (not shown) was made because at large sizes it is no longer beneficial to have more than one thread per particle, and the overhead of the code associated with summing over threads is noticeable. The neighbor-list is arranged in column-major order, i.e., the first neighbors of all particles are consecutive in memory, then the second neighbors, etc. This allows for efficient (coalesced) memory access.

Note the use of pre-fetching when reading from the neighbor-list; while the force contribution of neighbor $i$ is computed, the index of neighbor $i + 1$ is being read from the neighbor list.

Within the kernel a call is made to a function `fij` (not shown), which calculates the contribution to the pair force on the current particle from a neighbor particle. `fij` itself

| quantity | name in kernel | CUDA variable |
|---|---|---|
| Number of thread-blocks | NumBlocks | gridDim.x |
| Particles per block ($p_b$) | PPerBlock | blockDim.x |
| Threads per particle ($t_p$) | TPerPart | blockDim.y |
| Particle index within block | MyP | threadIdx.x |
| Thread index w.r.t. particle | MyT | threadIdx.y |
| Index of thread-block | MyB | blockIdx.x |
| Global index of particle | MyGP | MyP+MyB*PPerBlock |

Table 1: Short-hand notation for common quantities used in CUDA-kernels.

```
__global__ void Calcf_NBL_tp( ... )
  [ Declare shared memory ]
  float4 my_f = {0.0f, 0.0f, 0.0f, 0.0f};          // Initialize the force
  float4 my_r = LOAD(r[MyGP]);                      // Position of this particle
  int type_i = __float_as_int(my_r.w);             // Type of this particle
  [ Read information on the simulation box from device memory ]
  [ Copy potential parameters to shared memory ]
  __syncthreads();     // Parameters loaded to shared memory before proceeding

  int nb, my_num_nbrs = num_nbrs[MyGP];            // Read number of neighbors
  nb_prefetch = nbl[nvp*MyT + MyGP];               // Read first neighbor
  for (int i=MyT; i<my_num_nbrs; i+=TPerPart) {  // Loop over neighbors
    nb = nb_prefetch;
    if(i+TPerPart < my_num_nbrs)
      nb_prefetch = nbl[nvp*(i+TPerPart) + MyGP]; // Prefetch next neighbor
    float4 r_i = LOAD(r[nb]);                       // Read position of neighbor
    int type_i = __float_as_int(r_i.w);           // Type of neighbor
    // Add contribution from this pair to my_f:
    fij( potential, my_r, r_i, &my_f, [parameters, simulation box] );
  }
  __syncthreads();

  // Now use the shared memory for summing force contributions:
  s_r[MyP+MyT*PPerBlock] = my_f;
  __syncthreads();
  // Sum over threads associated with the same particle:
  if( MyT == 0 ) {
    for( int i=1; i < TPerPart; i++ ) my_f += s_r[MyP + i*PPerBlock];
    my_f.w *= 0.5f;    // Compensate for double counting of potential energy
    // Write result to device memory
    f[MyGP] = my_f;
  }
}
```

Figure 3: Kernel calculating forces on particles given the neighbor-list (nbl) shown in the simplest version where only the force and potential energy of each particle are computed. For a given particle each of $t_p$ threads (MyT = 0, 1, ..., $t_p$−1) computes part of the total force, which is summed up at the end. The function fij (not shown) adds an individual pair contribution to the current thread's force (my_f). Note the use of __syncthreads to synchronize threads within a thread-block. This is to ensure that all data are available in shared memory before any thread reads from it (first and third occurrences) or that all threads are done with the data in shared memory before it is used for other data (second occurrence). LOAD() is a macro that reads from device memory via the read-only data-cache using __ldg() on cards where this is available.

calls a function ComputeInteraction unique to each type of pair potential and selected via templating. Templating is used so that it is known when compiling fij which potential, and thus which ComputeInteraction, is to be called. Templating is also used for some of the other user-chosen variables, including the type of boundary conditions (represented by a SimulationBox class) and the cutoff-method. This means that the force calculation kernel is compiled for all possible combinations of these parameters, and the user can choose the appropriate one at run time. The code for the conditional statements which allows this is tedious, but is generated automatically by a Python script. The main disadvantage of using templating is that it increases the compile time considerably.

```
__global__ void calculateNBL_N2( ... ) {
  const unsigned int tid = MyP + MyT*PPerBlock;  // Thread-index within block
  [ Declare shared memory: s_r, s_Count, s_cut_skin2 ]

  if (updateRequired[0]) {
    if (MyT==0) s_Count[MyP]=0;      // Count of neighbors for this particle
    [ Copy cut-offs plus skin squared to shared memory]
    float4 my_r = r[MyGP];                    // Position of this particle

    // Loop over blocks of particles
    for (FirstGP=0; FirstGP<numParticles; FirstGP+=TPerPart*PPerBlock) {

      // Read particle positions in block into shared memory
      if (FirstGP + tid<numParticles) s_r[tid] = r[FirstGP + tid];
      __syncthreads();                        // Shared data in s_r ready

      // Loop over particles in block
      for (int i=0; i<PPerBlock*TPerPart; i+=TPerPart) {
        OtherP = i + MyT; OtherGP = FirstGP + OtherP;
        if (MyGP<numParticles && MyGP!=OtherGP && OtherGP < numParticles) {
          float4 r_i = s_r[OtherP];         // Position of other particles
          [ Read squared cutoff distance from shared memory based on types ]
          [ Calculate squared distance dist2 ]
          if (dist2 < RcutSk2) {
            // Atomically increment counter for this particle:
            unsigned int nextNbrIdx = atomicInc(&s_Count[MyP], numParticles);
            [ If space insert index into NB-list at position nextNbrIdx ]
          } // if(dist2 ... )
        } // if (MyGP ... )
      } // for(int i ... )
      __syncthreads();

    } // for (int firstGP ... )
    __syncthreads();

    if (MyT == 0) {
      [ Store the number of neighbors for this particle]
      [ Store its position so can check when rebuild needed ]
      [ Detect whether ran out of space and set flag to inform host ]
      if (MyP == 0) atomicDec(&(updateRequired[0]), NumBlocks);
    } // if (MyT == 0 ... )
  } // if(update_required[0])
}
```

Figure 4: Kernel for order-N$^2$ neighbor-list generation. Note that because the number of particles is not in general a multiple of $p_b$, there are some threads in the last block which should not do anything, hence statements such as if(MyGP<numParticles).

# 6 Neighbor-list generation: Order-$N^2$

This neighbor-list generating algorithm (see Fig. 4) has $O(N^2)$ complexity and is thus suitable only for small system sizes. In a serial code there would be a double loop; in a parallel code one loop (over particles whose neighbors are to be found) is handled completely by parallelization. Part of the loop over "other" particles is handled by looping over $t_p$-sized groups, while parallelization (the $t_p$ threads for that particle) accounts for looping within these groups (we do not make use of Newton's third law). Shared memory is used to reduce the amount of reads from device memory; in a straight forward implementation without shared memory a total of $N^2$ reads of particle positions is necessary. By using a block-wise reading into the shared memory, this is reduced to $N^2/p_b$, where $p_b$ is the number of particles in a block (denoted PPerBlock in the code). From this consideration $p_b$ should be as large as possible, but on the other hand a too large $p_b$ value would mean that the number of blocks ($\approx N/p_b$) becomes too small to utilize all the available SMX multiprocessors. RUMD uses the autotuner

```
__global__ void calculateNBL_CellsSorted( ... ) {
  gtid = blockIdx.x*blockDim.x + threadIdx.x; Count = 0;
  [ Declare shared memory: s_r, s_cut_skin2 ]
  [ Copy cut-offs plus skin squared to shared memory ]
  __syncthreads();
  if (gtid<numParticles) {
    float4 my_r = r[gtid];
    int3 my_CellCoordinates = calculateCellCoordinates(my_r, ...);
    int3 OtherCellCoordinates;

    // Loop over neighboring cells, applying periodic boundary conditions
    for (int dZ=-2; dZ<=2; dZ++) {
      OtherCellCoordinates.z = (my_CellCoordinates.z + dZ +
          num_cells_vec.z)%num_cells_vec.z;
      for (int dY=-2; dY<=2; dY++) {
        OtherCellCoordinates.y = (my_CellCoordinates.y + dY +
            num_cells_vec.y)%num_cells_vec.y;
        for (int dX=-2; dX<=2; dX++) {
          OtherCellCoordinates.x = (my_CellCoordinates.x + dX +
              num_cells_vec.x)%num_cells_vec.x;

          // Loop over particles in cell
          int otherCellIndex = calculateCellIndex(OtherCellCoordinates,
              num_cells_vec);
          int Start = cellStart[otherCellIndex];
          int End =   cellEnd[otherCellIndex];
          for (int OtherP=Start; OtherP<=End; OtherP++) {
            if (gtid != OtherP) {
              float4 r_i = LOAD(r[OtherP]);
              [ Read squared cutoff distance from shared memory based on
                 types ]
              [ Calculate squared distance dist2 ]
              if (dist2 < RcutSk2)
                [ If space insert index into NB-list and increment Count,
                   else break]
            }
          } // end for (int OtherP....)
        }
      }
    } // end for (int dZ ... )
    [ Store this particles number of neighbors]
    [ Store its position so can check when rebuild needed ]
    [ Detect whether ran out of space and set flag to inform host]
    if ( gtid==0 ) updateRequired[0] = 0;
  } // if(gtid < numParticles)
}
```

Figure 5: Kernel for order-N neighbor-list generation. `calculateCellCoordinates(...)` calculates the coordinates of the cell that a particle belongs to. `calculateCellIndex(..)` calculates the index of a cell given its coordinates. The arrays `cellStart` and `cellEnd` contain the indices of the first and last particles, respectively, associated with a given cell.

to pick the optimal value of $p_b$.

The kernel uses $t_p$ threads for a given particle to search for neighbors. This means that we have to deal with the situation that several or all of them find a neighbor at the same time, and the writing to the neighbor list should be performed without race-conditions. This is achieved by a so-called *atomic* operation. When several threads perform an atomic operation on the same variable, all operations are guaranteed to be performed in (an unspecified) sequential order. Here we use the atomic increment function, `atomicInc()`, which ensures that the number of neighbors is counted correctly. When a thread calls `atomicInc()`, the function returns the value the variable had *before* the increment of the given thread is performed. This is here used to specify a unique position in the neighbor list (`nextNbrIdx`).

The information about whether the neighbor-list needs to be rebuilt resides on the de-

vice, generated by a different kernel. The kernel in Fig. 4 is called at every timestep and checks via `if(updateRequired)` whether there is anything to be done. This is faster than copying the value of a flag to the host and having the host decide whether to launch the rebuild-kernel. `updateRequired` is initially equal to the number of thread-blocks. One thread from each block decrements it with an atomic operation (`atomicDec()`) when it (its thread-block) is done, so that when all blocks are finished, it is zero. At the next time step, assuming no particles have moved more than half the skin distance, `updateRequired` will still be zero and therefore the kernels immediately exit. Using an atomic operation to decrement `updateRequired` is necessary because the thread-blocks execute asynchronously, so none of them knows when/whether the others are finished, or even started; any unfinished blocks need to see a non-zero value of the counter.

The above means that for small systems the simulations are performed entirely on the GPU without any communication with the CPU (except when output is required). Avoiding the overhead associated with communication between the GPU and CPU is important for the performance at small system sizes.

## 7 Neighbor-list generation: Order-$N$

The order-$N$ algorithm is based on a cell-index method [1, 2] and involves (1) dividing the simulation box into rectangular spatial cells whose size is related to the potential cutoff; (2) associating particles with the appropriate cell based on the coordinates; (3) sorting the particles according to cell-index and rearranging all particle data to the sorted order (this can be done quickly with the Thrust library [55]). The advantage of rearranging the particle data to the sorted order is two-fold; i) the information about which particles are in a given cell can be stored simply as two integers indicating the first (`cellStart`) and the last particle (`cellEnd`); ii) better performance of the data-cache when reading the particle information both in the neighbor-list creation and in the force calculation.

The kernel in Fig. 5 is called after steps (1) to (3) have been carried out via a series of small kernels and Thrust operations. It involves, for a given particle, identifying its cell coordinates and looping over neighboring cells in three dimensions to find neighbors. We have chosen cell lengths in each direction to be of order (not less than) $(r_c + s)/2$, where $s$ is the neighbor-list skin. This means that the loop extends to plus/minus two cells in each direction, or 125 cells altogether. Such a choice of cell length means one searches a volume 58% [2] of that searched when using cells of length $r_c + s$. This kernel is called with one thread per particle, since that is generally most efficient at larger sizes, which is also when the linear method of neighbor-list generation becomes relevant. It is conceivable that some gain at intermediate sizes could be achieved by implementing a $t_p > 1$ version of the kernel, but this has not been tried yet.

In this Neighbor-list method the information about whether to rebuild the neighbor-list must be communicated to the host because several kernels and Thrust functions must be called (the use of Dynamic Parallelism, available since CUDA 5.0, could change this, but has not been tried). Thus the `updateRequired` flag is not used in the kernel because the kernel only runs at all if a rebuild is required; the flag is simply set to zero at the end by the thread handling particle 0.

---

[2]One must search for neighbors in a given particle's own cell and one (two) neighboring cells for cell length $r_c + s$ ($r_c + s/2$). In the former case the search volume is $27 (r_c + s)^3$; in the latter it is $125 ((r_c + s)/2)^3$ giving a ratio $(125/8)/27 = 0.58$.

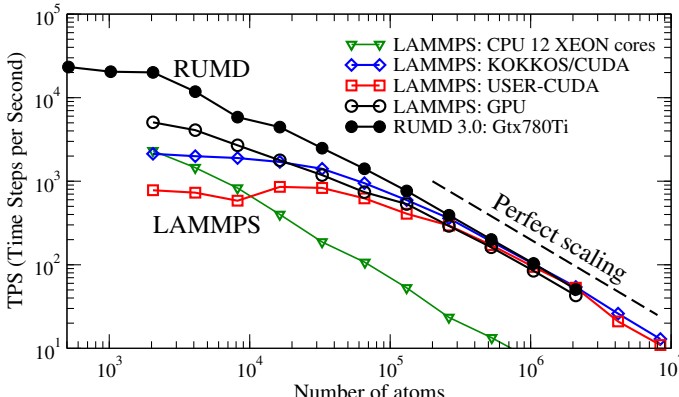

Figure 6: The Lennard-Jones LAMMPS benchmark: a melting FCC crystal is simulated at constant energy. LAMMPS results are downloaded from the LAMMPS webpage (see text). The vertical axis shows the number of simulated time steps per second of wall-clock time. At large system sizes all codes follows the ideal $1/N$ scaling, and the GPU's are 10-20 faster than LAMMPS running on 12 xeon cores. For RUMD good scaling is maintained down to quite small systems $N \sim 2000$, and at small system sizes RUMD is thus considerably faster than the three GPU versions of LAMMPS.

## 8 Benchmarks and performance analysis

To illustrate the performance of RUMD, we first compare to the "Lennard-Jones" benchmark results published on the LAMMPS homepage.[3] There are two reasons for this choice: i) the LAMMPS benchmark results include data for small systems (down to 2048 particles); ii) data is provided for three different GPU extensions of Lammps. The LAMMPS benchmark involves an FCC crystal of Lennard-Jones particles which is given a kinetic energy sufficient to melt it at constant total energy (NVE). Figure 6 shows as open symbols the number of timesteps per second (TPS) achieved by different versions of LAMMPS: A pure CPU version running on 12 Intel Xeon cores (dual hex-core 3.47 GHz Intel Xeons X5690), and three different GPU-extensions, KOKKOS/CUDA, USER-CUDA, and GPU, all running on a K20x card with 2688 cores. The corresponding results for RUMD running on the comparable Gtx780Ti card are plotted as filled circles. All the GPU-accelerated versions of LAMMPS, together with RUMD, give similar performance for large $N$ (above $\sim 3 \times 10^5$). In this regime near perfect scaling with $N$ is observed, and the GPU versions are 10-20 times faster than LAMMPS running on 12 Intel Xeon cores. At smaller system sizes the near perfect scaling breaks down: for two of the GPU versions of LAMMPS (the red and blue curves) running a simulation with 2000 particles takes as much time as one with 20000 particles; clearly the GPU hardware is under-utilized. In fact, for these two implementations it is faster to use the pure CPU version of LAMMPS at the smallest system sizes. RUMD, on the other hand, maintains reasonable (though not perfect) scaling down to around $N = 2000$. We have included even smaller system sizes, to illustrate that RUMD eventually also begins to struggle to utilize the hardware at very small system sizes.

Table 2 gives the parameters chosen by the autotuner, as a function of system size. Except for the two smallest system sizes, the autotuner chooses the total number of threads ($N \times t_p$) to be at least 16000. This illustrates the point made in the introduction, that the number of threads should be much larger than the number of physical cores (here 2880) to get good performance. The reason that fewer threads are used for the two smallest system sizes is

---

[3]http://lammps.sandia.gov/bench.html#gpucluster
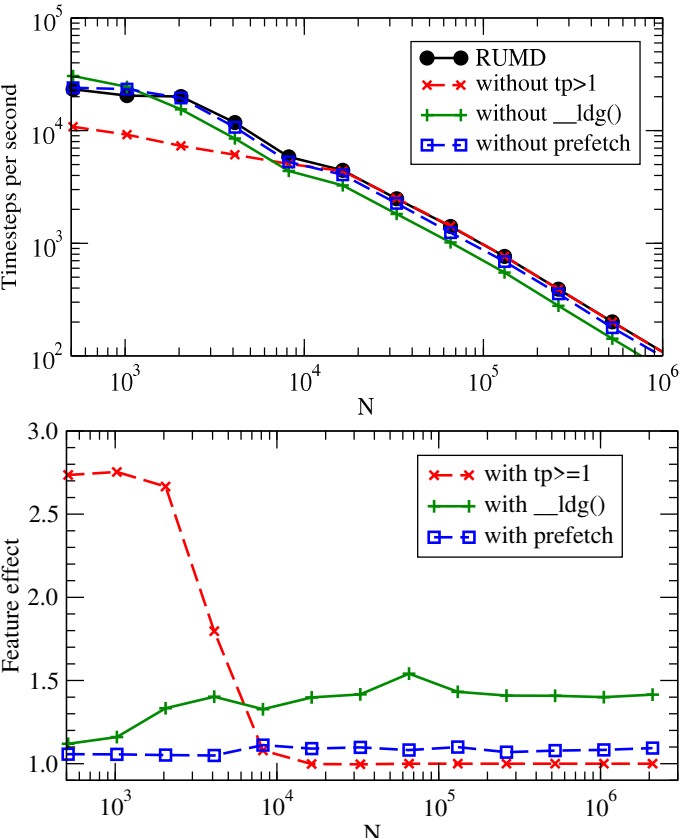

Figure 7: Analyzing the effect of optimization features in RUMD. The upper panel shows the performance of the full-RUMD and three other versions in which one feature has been disabled: multiple threads per particle ($t_p > 1$), use of read only data-cache to read positions (__ldg()), and pre-fetching. The lower panel shows the same data in terms of the relative boost in performance that each feature gives, as a function of system size.

| N | NB | pb | tp | skin | TPS |
|---:|:---:|---:|---:|---:|---:|
| 512 | $N^2$ | 16 | 14 | 0.452 | 31201 |
| 1024 | $N^2$ | 16 | 10 | 0.5 | 26729 |
| 2048 | $N^2$ | 48 | 8 | 0.611 | 20504 |
| 4096 | $N^2$ | 32 | 4 | 0.746 | 11222 |
| 8192 | $N$ | 192 | 2 | 0.824 | 6292 |
| 16384 | $N$ | 192 | 1 | 0.5 | 4772 |
| 32768 | $N$ | 192 | 1 | 0.5 | 2710 |
| 65536 | $N$ | 128 | 1 | 0.452 | 1458 |
| 131072 | $N$ | 192 | 1 | 0.409 | 771 |
| 262144 | $N$ | 128 | 1 | 0.409 | 382 |
| 524288 | $N$ | 128 | 1 | 0.370 | 194 |
| 1048576 | $N$ | 128 | 1 | 0.370 | 100 |
| 2097152 | $N$ | 96 | 1 | 0.335 | 50 |

Table 2: Performance parameters chosen by the autotuner and the resulting TPS (Timesteps Per Second) on a Gtx780Ti card (5.0 TFlops sp, 336GB/s).

| N | NB | pb | tp | skin | TPS | SpeedUp |
|---|---|---|---|---|---|---|
| 512 | $N^2$ | 16 | 4 | 0.335 | 35354 | 1.13 |
| 1024 | $N^2$ | 16 | 5 | 0.370 | 34226 | 1.28 |
| 2048 | $N^2$ | 64 | 8 | 0.5 | 31454 | 1.53 |
| 4096 | $N^2$ | 32 | 4 | 0.55 | 17508 | 1.56 |
| 8192 | $N$ | 32 | 1 | 0.611 | 8824 | 1.40 |
| 16384 | $N$ | 128 | 1 | 0.452 | 6617 | 1.3 |
| 32768 | $N$ | 32 | 1 | 0.553 | 3075 | 1.13 |
| 65536 | $N$ | 128 | 1 | 0.452 | 1798 | 1.23 |
| 131072 | $N$ | 192 | 1 | 0.409 | 948 | 1.23 |
| 262144 | $N$ | 192 | 1 | 0.409 | 495 | 1.30 |
| 524288 | $N$ | 128 | 1 | 0.370 | 239 | 1.23 |
| 1048576 | $N$ | 192 | 1 | 0.370 | 126 | 1.26 |
| 2097152 | $N$ | 192 | 1 | 0.370 | 58 | 1.16 |

Table 3: Performance parameters chosen by the autotuner and the resulting TPS (Timesteps Per Second) on a Gtx1080 card (8.2 TFlops sp, 320GB/s). Last column is speed-up compared to the Gtx 780Ti card (Table 2).

probably that the required large $t_p$ values inflict too large a penalty due to the sequential summation of the $t_p$ different contributions to the force (see Fig. 3). The switch between the two methods for neighbor-list generation happens at around 8000 particles. In this range of system sizes both methods are sub-optimal and the autotuner compensates by increasing the skin size to make neighbor-list updates less frequent.

Figure 7 shows the effect of disabling different optimization features. The upper panel shows time steps per second like Fig. 6, but with different curves representing different disabled features (the black curve is with all features enabled). The most dramatic difference is when $t_p = 1$ is enforced, for small and medium systems ($N < 10^4$). No difference is observed at larger $N$ because there $t_p = 1$ is the optimal choice, see table 2. Disabling the use of the read-only data-cache gives the green curve, a significant drop in performance across all sizes except the very smallest $N < 2000$, while disabling pre-fetching gives a slight drop, more at larger sizes. The lower panel of Fig. 7 shows the same data, but plotted as the ratio of the speed of the full RUMD to that of RUMD with the given feature disabled. Plotting this ratio, on a linear scale, shows the relative effects more clearly. In particular, reading via the read only data-cache gives an effect of order 40%, while pre-fetching has an effect of order 10% at the large sizes.

Table 3 gives results for RUMD running on a GTX 1080 card of the newer Pascal architecture. This card has a single precision theoretical peak-performance of 8.2 TFlops (8.9 TFlops in 'Boost' mode), and a memory bandwidth of 320 GB/s. These numbers are, respectively, 64% higher and 5% *lower* than the corresponding numbers for the Gtx 780Ti card. Table 3 shows speed-ups in the range 13 to 56%. The largest speed-ups are found at $N = 2048$ and $N = 4096$, where the $N^2$ Nb-list method is chosen – this method has more calculations per read from memory (see section VI), so the increased computational speed without increase in memory bandwidth can better be utilized. It should be noted that cards of the Pascal architecture with higher memory bandwidth have recently been released — our results suggest that these should perform even better at the large system sizes.

| N | $2 \cdot 10^3$ | $2 \cdot 10^4$ | $10^5$ |
|---|---|---|---|
| LAMMPS SP | 3.26 | 5.66 | 6.06 |
| LAMMPS WOLF | 2.21 | 3.57 | 3.40 |
| LAMMPS PPPM | 0.54 | 0.40 | 0.35 |
| RUMD SF | 9.41 | 11.3 | 16.1 |

Table 4: Benchmarks for a system of charged Lennard-Jones particles (see text for details) for LAMMPS (CPU) and RUMD (Gtx 780Ti). The metric shown is MATS (millions of atom time steps per second). LAMMPS benchmarks were performed on a Dell 630 server with dual Intel Xeon E5-2699 v3 2.3 GHz CPU's for a total of 36 cores. Coulomb interactions were evaluated using a simple shifted-potential cutoff at distance 6.0 ("SP"), using the Wolf method [56] with the switching parameter $\alpha$ set to zero ("WOLF", equivalent to the shifted force method), and the Particle-Particle Particle-Mesh method ("PPPM"). RUMD benchmarks were performed on an nVidia GTX 780 Ti, using a shifted force cut-off (SF) at distance 6.0.

## 9 Electrostatics

A general purpose MD code should include electrostatic (Coulomb) interactions and these should be sufficiently accurate and computationally efficient. Implementation of the smooth particle mesh Ewald method [21, 57–59] which can efficiently handle the long range part of the electrostatic interactions is planned, but for our needs so far we have found it sufficient to use Coulomb forces with a large shifted-force cut-off as documented in Ref. [60]. In that paper it was shown that a shifted-force cutoff of order five inter-particle spacings gives, similar to the Wolf [56] method, results in excellent agreement with Ewald-based methods in bulk systems.

To benchmark the performance of Coulomb interactions, we performed simulations of a model molten salt in RUMD and the CPU version of LAMMPS. All particles have identical Lennard-Jones parameters $\epsilon$ and $\sigma$. The charges are $\pm\sqrt{4\pi\epsilon_0\epsilon\sigma}$ (50% each). The density is $0.3677\ \sigma^{-3}$ and the temperature $2\ \epsilon/k_B$. The density is the same as was used by Hansen and McDonald in their study of a similar model salt [61]. In Ref. [60] it was shown that a cutoff of $6.0\sigma$, corresponding to $\sim 330$ neighbors per particle at this density, was sufficient to get satisfactory accuracy. The time step is $0.004\ \sqrt{m/\epsilon\sigma}$.

The data in Table 4 compare RUMD and the CPU-version of LAMMPS with different methods of evaluating Coulomb interactions. The benchmarks are expressed as MATS (million atom time steps per second) for ease of comparing different system sizes. The smallest speed-up of RUMD over LAMMPS is here a factor of two. This smaller speed-up compared to the previous section is primarily due to the LAMMPS benchmarks being run on a faster CPU system, a Dell 630 server with 36 XEON cores. For comparison, at the time of writing the cost of the Dell 630 server is roughly 10,000$, whereas the cost of the GTX 780 Ti card is around 500$ (to this should be added the cost of a fairly standard PC, which can hold three GPU cards).

## 10 Summary

We have described the RUMD software package for molecular dynamics simulation on GPUs, concentrating on the optimization strategies that distinguish it from most other GPU MD codes. We have documented its strong performance at small and medium system sizes and its performance comparable to other GPU-based MD codes at larger sizes. Work will continue on RUMD both with regard to features and optimization opportunities. The ability to split a simulation over multiple GPUs will also be considered, which will not just allow larger systems, but also even faster simulations of medium systems, given that RUMD already makes good use of the hardware for such sizes.

## 11 Appendix: The autotuner

Here we describe the algorithm used by the autotuner, which optimizes the choice of neighbor list algorithm, the neighbor-list skin size, and the way the generation of the neighbor list and the calculation of non-bonding forces are distributed among the GPU threads ($p_b$ and $t_p$). The basic strategy is to run a series of short simulations (a few hundred to a few thousand time steps) varying the different parameters, to find a set of of parameters giving (close to) optimal performance. Not all possible combinations of parameters are attempted in order to reduce the time taken for tuning (for Lennard-Jones-type systems without molecules or Coulomb interactions the tuning process takes under a minute for small systems, several minutes for larger systems). The initial state of the system is stored so that all comparisons made by the autotuner involve runs of the same length starting from the same configuration.

If the autotuner is not used, RUMD uses default values which depend on the system size: "n2" neighbor-list method (section 6) for $N < 8000$, otherwise "sort" (section 7); $p_b = 32, t_p = 4$ for $N \leq 10000$, otherwise $p_b = 64, t_p = 1$. The default skin value is 0.5, which assumes units of length such that the interparticle spacing is of order unity. In principle the default skin should be based on the interparticle spacing (e.g. $\rho^{-1/3}/2$), but in practice length units in MD are generally of order the interparticle spacing and the autotuner can quickly deal with a discrepancy. For some very small systems, $N < 200$, with not too small cutoff it can be faster not to use a neighbor-list at all. The autotuner checks this possibility for systems with $N < 5000$.

The dependence of performance on the parameters $p_b$, $t_p$ and neighbor-list skin is simple: the time taken shows a single minimum as a function of the parameter. This allows a relatively straightforward optimization strategy to be used. The number of steps run for the different stages depends on the system size (larger for smaller systems sizes to get better timing) and can be altered by the user but should not need to be. The overall strategy is as follows:

1. Run some steps before tuning (default: 10000) to avoid the influence of transient effects (associated for example with having changed the temperature).

2. Run with default parameters to get a baseline performance.

3. Phase I optimization: With the default $p_b$ and $t_p$ run with the different neighbor-list methods, "none", "n2", "sort". For each one the skin is optimized separately.

4. Phase II optimization: For the fastest neighbor-list method and other methods within 20% of the fastest, optimize the parameters $p_b$ and $t_p$ using a double loop: first $p_b$ considering the values 16, 32, 48, 64, 96, 128 and 192. For each $p_b$, values of $t_p$ are tested starting from 1 and increasing until 64. For each combination of $p_b$ and $t_p$ the skin length is re-optimized starting at the last identified optimal value.

5. If the neighbor-list method "n2" is included in phase II, it can still help to sort the particles once every few hundred times, typically for system sizes near the crossover from "n2" being optimal to "sort" being optimal. This is checked and the optimal sorting interval found.

6. If more than one neighbor-list method was optimized in phase II, make a final comparison between the phase II-optimized sorting methods to choose the overall optimized set of parameters (except close to the cross-over from one method to another, the phase II optimization does not change which method is chosen, and in that case the difference is small anyway).

7. Run, using the optimized parameters, the same number of steps as for the baseline run to determine the overall improvement due to tuning.

8. Write the tuned parameters to a file so that repeating the simulation in the same directory with the same "user parameters" does not require re-tuning. For this purpose, "same user parameters" means: same number of particles of each type, same density and temperature and potential parameters (within a tolerance), same integrator type and timestep (within tolerance), and same GPU-type. The actual configuration does not have to be the same. If there is any doubt about re-using the previously found parameters, the file can just be deleted.

Some further details are noted here:

- The skin optimization starts from the default value (phase I) or previously identified phase I-optimal value (phase II). Its value is increased and decreased in steps of 20% (phase I) or 10% (phase II) until a minimum is identified in the time taken. Attempting to optimize the skin to a precision of better than 10% is not worth the effort.

- The loop over $t_p$ breaks out when one of the following three conditions is met: (i) the time taken exceeds the minimum time so far three times in a row; (ii) the time taken exceeds the minimum time so far by 5%; (iii) some GPU resource-limit is exceeded, either the number of threads per block ($p_b t_p$) or the total register count per block.

- The loop over $p_b$ breaks out when the time taken (having optimized $t_p$ and skin) exceeds the previous best by 10% or more.

- For very large systems it doesn't make sense to use anything other than the "sort" method for the neighbor-list. The autotuner omits "none" for $N > 5000$ and "n2" for $N > 50000$. Moreover, large systems generally require larger $p_b$ and so the autotuner omits $p_b = 16$ for $N > 4000$ and $p_b = 32$ for $N > 10^5$. Also for the largest systems only $t_p = 1$ is relevant; the autotuner omits checking other values for $N > 10^5$.

## Acknowledgements

A large part of this work was sponsored by the Danish National Research Foundation's grant DNRF61. This work was supported in part by the VILLUM Foundation's grant "Matter".

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
