# Peer review of "RUMD: A general purpose molecular dynamics package optimized to utilize GPU hardware down to a few thousand particles"

_SciPost Physics, doi:SciPost Phys. 3, 038 (2017)_

## Round 3 · Referee Report · Anonymous · 2017-10-22

Strengths
1) Introduces an efficient package for MD simulations on GPUs, which does not waste hardware resources for small to medium size systems. This offers in particular the possibility to simulate systems with 10^3-10^4 particles over very long time scales, e.g. to investigate glass-forming liquids
2) Nice overview of the features offered by the package, presentation of the main optimization strategies
Weaknesses
1) Even though the manuscript reads well, it may look too technical to the non-specialized reader
Report
The manuscript submitted by Bailey et al. reports on a molecular dynamics package, RUMD, optimized to efficiently use GPU resources not only for the usual goal of large systems, but also for smaller systems down to a few thousands of particles. Such an objective may sound counter-intuitive, but its relevance is well justified by the authors, who put forward three main reasons for targeting such performance: 1) simulating long time scales rather than large systems (this is particularly relevant for glass-forming liquids — a field in which the authors are well-known experts — and can be readily applied using the currently available hardware and the RUMD package) 2) as a building-block for multi-GPU simulations (which is indicated as a perspective not dealt with in this manuscript) 3) anticipating hardware development, with increased number of cores, from which the present work should benefit automatically.
Even though quite technical, the manuscript reads well. It does not describe the basics of GPU programming, but rather insists on the features which make the implementation of RUMD particularly efficient for small systems, in particular the optimization strategies which allow to limit the under-utilization of the hardware. Of particular interest is the use (and description in the appendix) of the autotuner, which optimizes the relevant parameters for a given system: neighbor list algorithm, associated skin size, and distribution among GPU threads of the neighbor list generation and force calculation. The performance of RUMD for Lennard-Jones and charged Lennard-Jones fluids is demonstrated and compared to various implementations of the LAMMPS code. Compared to the latter, better performance is achieved on GPUs for small to medium sized systems, while keeping a similar performance for large systems.
I recommend this manuscript for publication after the authors have addressed the following points.
Requested changes
1) The performance is measured with respect to that of the LAMMPS code and its implementations on GPU, for the same number of cores (2688), as well as a pure CPU version running on 12 Xeon cores — for which a weaker performance is found. Could the authors explain this choice of reference for CPUs, since the number of cores is so different?
2) In the same spirit, the authors mention in the introduction several existing MD codes based on GPUs, but no comparison is then provided to measure the relative performance of RUMD. Even though such a numerical comparison may be out of the scope of the present work, could they explain what are the specific features of RUMD which resulted in their choice of developing a new code instead of using the existing ones? In addition, it might be useful in the introduction to explicitly mention why GPUs might be interesting for MD in general, so that non-specialist readers may also appreciate this point.
3) A list of available features is provided. The authors also mention the project of implementing Ewald summation. Indeed I believe that this is an important point, since this approach is more standard than the truncated-shifted/Wolf method. Could the authors explain how this could be achieved? Another important feature (which I believe is not yet included, but I might have missed it) is that of constraints on bonds/angles, as necessary e.g. for the most standard water models.
4) On page 6, the authors refer to « devices of compute capability at least 3.5 ». This is unclear to me. Could the authors explain what this means?
5) The algorithm presented in Figure 3 (and in the text in general) discusses only pair interactions. Is there a different implementation for more-than-two-body interactions? In the same figure, the line referring to double-counting deals with the variable my_f.w Could the authors explain what this is?
6) On page 9, « aviable » probably means « available »?
7) On page 11, could the authors explain further the calculation leading to the 58% estimate with the choice of cell size?
8) In Table 2 one observes a non-monotonic decay of the parameter pb in the range N 32768-262144 (associated with a decrease in the skin but at constant tp=1). Could the authors comment on this evolution?
Author: Nicholas Bailey on 2017-11-14 [id 189]
(in reply to Report 1 on 2017-10-22)
We thank the referee for his/her careful reading of the manuscript. Below we address the issues raised point by point, mentioning the changes in our resubmitted version.
Requested changes
1) The benchmarks for CPU were taken from the LAMMPS website; so we didn't explicitly make choose what CPU configuration to compare to. If the reviewer is asking why compare at all to a CPU-based MD implementation, it is simply a convenient reference for those who may not be familiar with what GPUs have to offer. It is desirable to have a certain speed-up over CPUs given the extra complexity of developing GPU-codes. In such comparisons it is important to compare with a state-of-the-art CPU-based implementation running on a modern CPU-configuration.
2) It is indeed beyond the scope of this paper to do a exhaustive investigation of small system performance of GPU based codes. We chose to compare to Lammps for two reasons: i) they publish performance data on their web-page including data for small systems (down to 2048 particles). ii) they provide data for three different GPU extensions of Lammps. The choice of developing our own code was tied to our interest in small systems, whereas the existing codes at the time were focused on larger systems and therefore had included little optimization for smaller systems. We have tried to emphasize this point already in the introduction (not to mention the title) (and from the reviewer's report it seems he/she has understood this point). Regarding mentioning the usefulness of GPUs for MD we have added a sentence in the second paragraph of the introduction: {\bf MD is a good candidate for GPU-acceleration, as discussed by Stone {\em et al.}, since it involves a reasonably high arithmetic intensity, that is number of floating point operations per memory access.}
3) The Ewald method will be the smooth particle-mesh Ewald method. We have not yet begun to think in detail about how to implement Ewald summation, but while non-trivial, it has been done more than once for NVIDIA GPUs. The text has been updated to refer explicitly to the smooth particle Ewald method rather than an Ewald-based method'' and relevant references have been included. As mentioned in the feature-list under
Other interactions'', we have included both rigid constraints and flexible intra-molecular bonding potentials for bonds, angles and dihedral angles. This point is been expanded slightly and two references for the constraint method included.
4) This is a term that NVIDIA uses to specify the technical specifications of different devices. We have modified the sentence to make it clear that this is an NVIDIA-specific term.
5) We emphasized pair potentials in the text because this is the most important case. Other kinds of potentials have been implemented in the software, but we feel it is not so useful to describe them in detail in the manuscript. For example bonding potentials (bonds, angles dihedral angles) are necessary for making flexible but implementing them in parallel is relatively straightforward, and their cost is typically small compared to non-bonding potentials. One kind of many-body potential, known as the effective-medium theory, has been implemented. Although it is many-body, it can be computed using the same kind of combination of looping and parallelism that we use for the pair potentials, except that the loop over neighbors in the force calculation must be done twice (the neighbor-list is the same). The performance is not much slower than for pair potentials. We are interested in implementing the more commonly used EAM many-body method for metals, but it is less clear how to do this on GPUs. We have also implemented a three-body potential, the so-called Axilrod-Teller potential for Argon. This requires an extra nested loop over neighbors and is therefore computationally much more demanding than the pair potentials.
As noted in the text, we do not make use of Newton's third law (the factor of two saved in floating point operations would be outweighed by the extra memory access and associated synchronization). The function \verb|fij| calculates a pair force and pair potential energy according to the specific potential object. Since this is called separately for both particles of a given pair, when adding the energy to each particle's total potential energy we divide by two, otherwise that pair energy will end up being counted twice. This is the standard way to assign potential energy to particles when there is a pair interaction. The variable used here, \verb|my_f|, is a float4, which has four components: \verb|my_f.x|, \verb|my_f.y| and \verb|my_f.z| hold the corresponding components of force on the given particle, while \verb|my_f.w| holds its potential energy.
6) Yes. It has been fixed.
7) An explanation has been included as a footnote.
8) First of all the value 196 is incorrect (typographical error); this should have been 192. It is important to note that at these large system sizes details such $pb$ matter less and less because it is easy to utilize the hardware efficiently (this is consistent with our code's performance matching the others we compared at larger sizes). The actual times found by the autotuner differ very little between these different values of $pb$. While the differences in optimal $pb$ are reproducible (so it is not due to timing fluctuations) and apparently systematic, the actual timing differences are small enough that we feel it is not worth the effort to explain the trend.
Author: Nicholas Bailey on 2017-11-14 [id 190]
(in reply to Report 2 by Daniele Coslovich on 2017-10-27)We thank Daniele Coslovich for his very positive review. We have of course fixed the typo he noticed.

---

## Round 3 · Referee Report · Daniele Coslovich · 2017-10-27

Strengths
1- The paper demonstrates convincingly the efficiency of the RUMD simulation package for small and medium size particle systems
2- It provides a succinct but complete description of the code internals
Weaknesses
No major weakness
Report
The manuscript describes the RUMD simulation package. It is a well-tested molecular dynamics code running entirely on graphic processing units (GPU). It distinguishes itself from related software for the very good performance on small and medium systems size, down to a few thousands particles and even less. This is achieved by dynamically adjusting the number of threads per particle involved in the force calculation at the beginning of the simulation. The paper describes succinctly this and other low level optimizations that enable very good efficiency on both small/medium and large system sizes. The direct comparison between RUMD and the well-established LAMMPS package for a standard Lennard-Jones benchmark nicely demonstrates the effectiveness of the approach. I have used RUMD myself in a few projects of mine and I confirm it runs efficiently on small systems, making it an ideal building block of more complex simulation strategies.
The paper is carefully written, with a sufficient level of detail about the code internals. Its publication on a peer-reviewed journal is long overdue and it is great to see it submitted to SciPost. I definitely recommend its publication.
Requested changes
Typos:
- Table 4: Lennard-Jones (not Lennard-Jone)

---

## Round 4 · Author Response

In this revised version of the manuscript we have made several minor changes in response to the first referee's comments. Some of the questions were answered in the response and we did not feel it necessary to change the manuscript (since the response is also public).

---

## Round 4 · List of Changes

1. Removed present address for one author. Several authors are no longer in academia so their present addresses are not relevant. We decided it looked better to only include people's Roskilde University affiliation (where they did the work).

2. Email address for last author TBS added.

3. First paragraph: sentence about usefulness of GPUs for MD simulation added

4. Reference added, see last sentence of third paragraph.

5. Keyword arguments (i, j) added to the example script in Figure 1 for consistency

6. References added to "Other interactions" item on Feature list, which has also been reworded slightly.

7. Clarification of the term "compute capability" in point 3 of optimization strategy list.

8. Comment added in the code in Figure 3. Some further simplication made by ommitting some lines towards the end of the function.

9. Added figure reference to first sentence in section 6 on neighborlist (order N^2)

10. Explained the 58% in section 7 in a footnote

11 Section 8 second sentence, added word "choice"

12 Fixed an incorrect value in Table 2 (should have been 192 not 196)

13 Elaborated slightly on which Ewald method we intend to implement, and included references to other GPU implementations of it.

14 Figure 7 the word "without" in the legend of Figure 7 (b) has been replaced with "with", which is more appropriate.

15 Missing s in Lennard-Jones added as noticed by second referee

---

## Editorial Decision

published